# Hemogram-Based Phenotypes of the Immune Response and Coagulopathy in Blunt Thoracic Trauma

**DOI:** 10.3390/jpm14121168

**Published:** 2024-12-21

**Authors:** Alexandru Emil Băetu, Liliana Elena Mirea, Cristian Cobilinschi, Ioana Cristina Grințescu, Ioana Marina Grințescu

**Affiliations:** 1Department of Anesthesiology and Intensive Care II, Carol Davila University of Medicine and Pharmacy, 050474 Bucharest, Romania; alexandru.baetu@rez.umfcd.ro (A.E.B.); ioana.grintescu@umfcd.ro (I.M.G.); 2Department of Anesthesiology and Intensive Care, Grigore Alexandrescu Clinical Emergency Hospital for Children, 011743 Bucharest, Romania; 3Department of Anesthesiology and Intensive Care, Clinical Emergency Hospital Bucharest, 014461 Bucharest, Romania; 4Department of Anesthesiology and Intensive Care, Zetta Clinic, 020311 Bucharest, Romania; ioana_grintescu@yahoo.com

**Keywords:** blunt thoracic trauma, neutrophil-to-lymphocyte × platelet ratio (NLPR), trauma-induced coagulopathy

## Abstract

**Background**: Blunt thoracic trauma possesses unique physiopathological traits due to the complex interaction of immune and coagulation systems in the lung tissue. Hemogram-based ratios such as neutrophil-to-lymphocyte (NLR), platelet-to-lymphocyte (PLR), neutrophil-to-lymphocyte × platelet (NLPR) ratios have been studied as proxies for immune dysregulation and survival in trauma. We hypothesized that blunt thoracic trauma patients exhibit distinct patterns of coagulation and inflammation abnormalities identifiable by the use of readily available hemogram-derived markers. **Methods**: The present study represents a retrospective observational analysis that included 86 patients with blunt thoracic trauma from a single high-volume level one trauma center. The primary outcome was mortality prediction in blunt thoracic trauma patients using these derived biomarkers. Secondary outcomes included phenotypes of the immune response and coagulopathy and the prediction of non-fatal adverse events. **Results**: A U-shaped distribution of mortality was found, with high rates of early deaths in patients with an NLPR value of <3.1 and high rates of late deaths in patients with NLPR > 9.5. A subgroup of blunt thoracic trauma patients expressing moderate inflammation and inflammation-induced hypercoagulation objectified as NLPR between 3.1 and 9.5 may have a survival benefit (*p* < 0.0001). The NLPR cut-off for predicting early deaths and the need for massive transfusion was 3.1 (sensitivity = 80.00% and specificity = 71.05%). **Conclusions**: These findings suggest that blunt thoracic trauma patients exhibit distinct phenotypes of the immune response and coagulopathy from the early stages. A controlled, balanced interaction of immune, coagulation, and fibrinolytic systems might effectively achieve tissue repair and increase survival in thoracic trauma patients and should be subject to further research.

## 1. Introduction

Global health policies, together with a better understanding of the physiopathology of traumatic injury, have led to the implementation of international evidence-based clinical guidelines for the management of trauma patients, ultimately contributing to a steady decline in mortality attributable to injury in the last few decades [1,2]. Despite this, European studies have shown that 15 to 43% of trauma deaths are still definitely or potentially preventable. The thorax is reported to be the main injured body region associated with preventable deaths, and blunt thoracic trauma is the primary injury mechanism [3,4].

Trunkey described a classic trimodal distribution of trauma deaths and consists of immediate deaths, from severe traumatic brain injury (TBI), or severe injury to the heart or great vessels; early deaths, mainly caused by significant hemorrhage; and late deaths, due to sepsis or multisystem organ failure. Overall, advances in trauma management have led to a decrease in early deaths, with a reduction in the historical third peak of late deaths [5,6].

Trauma-induced coagulopathy (TIC) remains the leading cause of preventable death associated with injury, either by early massive, uncontrolled bleeding or late thrombo-inflammatory complications. Clinical phenotypes of hemostasis reflect the dynamic balance between the four primary mechanisms of hypocoagulability: hyperfibrinolysis, fibrinogen depletion, platelet dysfunction, and decreased thrombin generation and hypercoagulability: namely, fibrinolysis shutdown, hyperfibrinogenemia, platelet activation, and increased thrombin generation, respectively [7].

Immune system activation is an immediate response to the release of damage-associated molecular patterns (DAMPs) from injured tissue. It contributes to initiating defense and initial hemostatic mechanisms through complex interactions with endothelial cells and platelets. However, a dysregulated exaggerated pro-inflammatory response eventually leads to endotheliopathy, clot formation and clot lysis abnormalities, and innate immune system dysfunction, which depicts the systemic inflammatory response syndrome (SIRS). Simultaneously, a counterbalanced response to SIRS through the compensatory anti-inflammatory response syndrome (CARS) aims to reestablish immune homeostasis by inhibitory mechanisms. If CARS is prolonged, its mechanisms also become maladaptive, leading to immunosuppression, delayed tissue reparation, susceptibility to infections, and multiple organ dysfunction syndrome (MODS) [8,9,10,11]. DAMPs are also activators of the complement system. They produce a rapid generation of C3a and C5a, followed by consumption, which eventually leads to an imbalance in the components of the complement cascade and immunosuppression or “trauma-induced complementopathy” [11,12]. The coagulation and the complement systems seem to be intertwined in the initiation and progression of SIRS and MODS after trauma [13,14].

Genome-wide analysis and cellular analysis, along with cytokine and complement profiling, have been used to characterize the interplay of coagulation and inflammation. However, they are of limited use to the clinician since they are not readily available at the bedside [8,12,15,16]. Both pro-inflammatory and anti-inflammatory cytokines have been involved in post-traumatic inflammatory responses and their use as severity, evolution, and mortality predictors in trauma patients has been extensively studied [8,17,18,19]. Xiao’s group demonstrated that the genomic response to trauma is similar to the changes in gene expression caused by severe burn injury or endotoxemia, introducing the concept of “genomic storm”, which suggests the extent to which the leukocyte transcriptome changes its expression patterns in response to severe trauma, similar to what happens in a “cytokine storm” [20,21].

Hemogram-derived ratios such as the neutrophil-to-lymphocyte (NLR), platelet-to-lymphocyte (PLR), and neutrophil-to-lymphocyte x platelet (NLPR)—defined as neutrophil/(platelet × lymphocyte count)—ratios have been studied as proxies of immune dysregulation and survival in multiple conditions, including multiple trauma, TBI, TBI-induced coagulopathy, and acute respiratory distress syndrome (ARDS) [22,23,24,25,26,27]. Studies investigating the NLR at admission in blunt thoracic trauma have evaluated its predictive value for delayed ARDS or pneumothorax development [28,29]. To our knowledge, no studies have investigated the role of any of these hemogram-derived markers in predicting specific TIC phenotypes in thoracic trauma patients.

We hypothesized that thoracic trauma patients exhibit clinically relevant patterns of coagulation and inflammation abnormalities from the early stages and that easily accessible, inexpensive hemogram-derived markers could promptly reflect this. We aimed to investigate whether there might be a subgroup of patients for whom inflammation-induced coagulation might bring a survival benefit and whether this could represent the basis for future personalized therapeutic approaches.

## 2. Materials and Methods

### 2.1. Population and Study Design

A retrospective observational study was conducted on all patients admitted to the Bucharest Emergency Clinical Hospital between April 2019 and March 2024 for isolated blunt thoracic trauma. The study was conducted in a high-volume level-one urban adult trauma center, in agreement with the Declaration of Helsinki and following approval by the Ethics Committee for the collection and analysis of data and result publishing under protocol code no. 11098/03.12.2019.

The Abbreviated Injury Scale (AIS) score was used to identify patients with isolated thoracic trauma. AIS represents a widely used, anatomically based trauma severity scoring system that classifies traumatic lesions by body region according to their relative severity on a 6-point scale (with 1 representing minor and 6 representing virtually unsurvivable lesions). This study included patients with an AIS score of between 3 and 5 for the worst blunt thoracic injury and equal to or less than 2 in any other two body systems [30].

The following exclusion criteria were considered: age under 18 years, multiple trauma involving thoracic trauma; burn injury; post-cardio-pulmonary resuscitation status; pregnancy; chronic anticoagulant, antiplatelet, or immunomodulatory treatment; chronic lung, liver or kidney disease; known malignancy; known autoimmune disease; patients in which rotational thromboelastometry (ROTEM Sigma) analysis was not performed; and patients transferred from other hospital units to our center later than six hours after the inflicting traumatic event. We enrolled *n* = 86 patients with isolated blunt thoracic trauma in our study (Figure 1).

### 2.2. Data Collection, Definitions, and Parameters

The data we used was derived from the first set of complete blood work analysis performed at admission to the emergency department (hemogram, classic coagulation assay, and ROTEM Sigma).

The following devices were used for blood work analysis: ROTEM Sigma rotational thromboelastometer, RAPIDPoint^®^ 500 Blood Gas Systems (Siemens Healthineers, Erlangen, Germany), Celltac F Automated Hematology Analyzer (Nihon Kohden, Tokyo, Japan), ARCHITECT c4000 (Abbott, Lake Forest, IL, USA), and ACL TOP 500 (Werfen, Bedford, MA, USA). Quality assurance and controls for monitoring the fidelity and precision of the tests were performed according to the device manufacturer’s guidelines.

In ROTEM Sigma, a whole blood sample is inserted into a cup in which a pin is suspended. The cup is not mobile, but the pin oscillates; thus, clot formation will determine a rotatory movement of the weldment based on which of the dynamics of clot formation, strength, and lysis are further analyzed. The viscoelastic assay uses channels with different clotting activating agents: EXTEM provides information similar to that of prothrombin time by using recombinant tissue factor; INTEM is similar to aPTT, using phospholipid and ellagic acid; FIBTEM, when compared with EXTEM analysis, allows for the qualitative assessment of the fibrinogen contribution to clot strength independently of thrombocytes—it uses cytochalasin D, recombinant tissue factor, and polybrene; APTEM, when compared with EXTEM analysis, evaluates fibrinolysis, using a serine protease inhibitor or a synthetic lysine analogue and recombinant tissue factor. All channels also contain calcium ions [31].

Demographical data, which include age, gender, weight, and body mass index, were collected and analyzed. Additionally, laboratory values such as pH; base excess (BE); serum lactate; hemoglobin levels (Hb); leukocyte (Leu), neutrophil (Neu), lymphocyte (Lym), and platelet (Plt) counts; CRP; fibrinogen (Fbg); international normalized ratio (INR); activated partial thromboplastin time (aPTT); and thromboelastometry parameters—the EXTEM channel (clotting time—CT; clot amplitude at 5 min—A5; maximum clot firmness—MCF—or maximum amplitude; maximum lysis—ML) and the FIBTEM channel (A5, MCF, ML)—were also analyzed.

We calculated the hemogram-derived markers of immune dysregulation as follows: NLR = Neu (×10^9^ cells/L)/Lym (×10^9^ cells/L), PLR = Plt (×10^9^ cells/L)/Lym (×10^9^ cells/L), and NLPR = Neu (×10^9^ cells/L)/(Lym (×10^9^ cells/L) × Plt (×10^9^ cells/L)). 

Reference ranges for ROTEM Sigma parameters were defined according to thromboelastometry-guided management algorithms in trauma as follows: hypercoagulability: **increased thrombin generation**—EXTEM CT < 45 s or EXTEM MCF > 68 mm, **hyperfibrinogenemia**—FIBTEM MCF > 22 mm, **fibrinolysis shutdown**—EXTEM ML < 3%; and hypocoagulopathy: **hyperfibrinolysis**—EXTEM ML ≥ 15% or FIBTEM ML ≥ 10%, **fibrinogen deficit**—EXTEM A5 < 35 mm or FIBTEM MCF < 12 mm or FIBTEM A5 < 9 mm, and **thrombin generation deficit**—EXTEM CT > 80 s [32,33,34].

Qualitative platelet deficits could be investigated by subtracting the FIBTEM MCF from the EXTEM MCF, thus assessing only the thrombocyte contribution to clot formation. However, the study of stress and strain and the result of their action at a molecular level, i.e., creep, shows the viscosity alterations and reflects a nonlinear relationship between clot amplitude and elasticity. This prompted us to use PLTEM, a derived parameter that uses clot elasticity to measure the force with which the clot opposes the pin transduction-system-induced oscillations. In trauma patients, correlational analysis showed a statistical significance between PLTEM and the thrombocyte count [35,36].

PLTEM requires the measurement of maximum clot elasticity (MCE) and is calculated as follows: PLTEM = EXTEM MCE − FIBTEM MCE. MCE is defined as (MCF × 100)/(100 − MCF) [36].

### 2.3. Follow-Up

All patients were managed in the ICU as per the latest guidelines on Advanced Trauma Life Support (ATLS) and Management of major bleeding and coagulopathy following trauma, with a focus on addressing specific chest trauma issues [2,37].

Patients were managed according to liberal blood transfusion principles until the results of the first viscoelastic assay were obtained. Afterward, they received blood products according to a goal-directed protocol. We documented whether patients received tranexamic acid in the prehospital setting.

Fluid resuscitation was initiated using crystalloids. Dynamic fluid responsiveness and tolerance evaluations guided the addition of vasopressor and inotropic support. Minimally invasive hemodynamic monitoring, transthoracic echocardiography, and transesophageal echocardiography in selected cases guided the maintenance of adequate cardiac output. The acid–base derangements were addressed promptly; normothermia was rapidly reached and maintained.

Intubated patients were mechanically ventilated using Dräger Evita^®^ V800 ventilators (Drägerwerk AG & Co., Lübeck, Germany), with tidal volumes of 6–8 mL/kg predicted body weight in a pressure-controlled mode. Positive end-expiratory pressure and -inspired oxygen fractions were titrated according to ARDS-net mechanical ventilation guidelines. Bronchoscopy was performed whenever there were radiological signs of atelectasis by blood and mucous plugs bronchial occlusion, and chest drains were placed for pleural disruptions resulting in clinically significant pneumo-, hemo-, or hemopneumothorax, as per ATLS guidelines [37].

### 2.4. Study Outcomes

The primary outcome used was mortality prediction in blunt thoracic trauma patients using NLR/PLR/NLPR/PCR. Secondary outcomes included phenotypes of the immune response and coagulopathy and the prediction of non-fatal adverse events.

### 2.5. Statistical Analysis

All data were collected manually and stored in Microsoft Office Excel, version 2021. GraphPad version 10.2. (GraphPad Software, Boston, MA, USA) software was used for statistical analysis. The D’Agostino–Pearson test was used for data normality assessment. The *t*-test (parametrical data) and Mann–Whitney test (non-parametrical data) were used to compare independent variables divided by the clinical outcome. We used ROC curves that had death at 30 days as the classification variable to compare the predictive ability of NLR, NLPR, PLR, and CRP.

The analysis of ROC curves was also used to derive the inflammatory profiles that appear following blunt chest trauma. The first outcome used was early deaths (within the first 72 h) or patients’ need for massive transfusion at hospital admission. After obtaining the cut-off point by the maximum Youden index for the NLPR, these patients were excluded, and we performed another ROC curve. The endpoint for this ROC curve was mortality at 30 days, since we wanted to integrate the late deaths caused by thromboembolic and infectious complications into the analysis. After generating a new cut-off point for the NLPR score, three groups were outlined according to the level of inflammation. Stettler et al. also used this statistical analysis model for defining fibrinolytic phenotypes in multiple trauma [38].

The survival of the three groups determined by the inflammatory profile following blunt chest trauma was compared using Kaplan–Meier curves (log–rank test). The paraclinical data of the three groups were compared using one-way ANOVA for parametrical data and the Kruskal–Wallis test for non-parametrical data. Analysis of distribution frequencies was performed using Fischer’s exact test. A *p* value < 0.05 was considered significant for all double-tailed tests.

## 3. Results

This retrospective observational study includes 86 patients, the majority of whom were men. The mean age of the population was 44.36 years (±15.14 years). Traffic accidents (*n* = 38; 44.18%) were the principal mechanism that led to chest trauma, followed by interpersonal physical aggression (*n* = 27; 31.39%), falls from a height (*n* = 18; 20.94%), and sports accidents (*n* = 3; 3.49%). The clinical and imaging examination of these patients revealed multiple specific injuries that were often combined: rib fractures (*n* = 54), flail chest (*n* = 9), small/large pneumothorax (*n* = 11/*n* = 6), minor/major hemothorax (*n* = 8/*n* = 5), chest wall contusion (*n* = 18), sternal fracture (*n* = 10), limited/extensive pulmonary contusion (*n* = 31/*n* = 16), myocardial contusion (*n* = 6), aortic injury–intramural hematoma/pseudoaneurysm (*n* = 2/*n* = 1), diaphragmatic rupture (*n* = 1), and esophageal rupture (*n* = 1).

Surgical management was needed in 27.90% (*n* = 24) of the cases. Distribution frequency analysis of surgical or conservative management showed no statistically significant difference in the survivor and non-survivor groups (Table 1).

Intercomparison of the groups divided after survival at 30 days revealed similar values for mean age, BMI, and chest AIS in the survivors and non-survivors. All acid–base markers of maximum importance revealed statistically significant differences except for serum bicarbonate. The base excess was higher in survivors, while serum lactate was higher in patients who died within the first 30 days, and serum pH followed the trauma-imposed metabolic trend. Patients who died were, overall, more acidotic than patients who survived (*p* < 0.0001). The hemoglobin level was higher among the surviving patients; although small, the difference between the median values was statistically significant (*p* = 0.02). Patients who died were more thrombocytopenic (*p* < 0.0001). The classic coagulation assay showed a prolonged INR in deceased patients (*p* = 0.003), but this difference was not supported by a corresponding increase in the aPTT value (*p* = 0.059). Given the trend toward hypocoagulability, obtaining a similar average value for the clotting time on the EXTEM channel (63.25 ± 21.11 vs. 66.27 ± 14.95) is surprising. However, although the initiation of coagulation appears similar in time, the lower clot amplitude at 5 min and maximum clot firmness in deceased patients seems to point to a poorer quality clot (Table 1).

All three hemogram-derived immune dysregulation markers have statistically significantly lower mean/median values in the non-survivor group (NLR: 7.86 ± 3.48 vs. 10.78 ± 5.61, *p* = 0.004; PLR: 0.14 IQR (0.10 to 0.19) vs. 0.19 IQR (0.13 to 0.29), *p* = 0.03; NLPR: 4.31 ± 2.51 vs. 10.66 ± 6.63, *p* < 0.0001).

CRP, a classic inflammation biomarker, has been extensively studied in trauma patients [39]. Its predictive value for mortality or adverse events is highly time-dependent, with a detection time of 6 to 8 h after the initial stimulation and a peak at 24 to 48 h [40]. This time-sensitivity underscores the importance of careful timing in its analysis, as its effectiveness as a mortality or adverse-events predictor is strictly contingent on the time of analysis, making it ineffective at admission. High CRP levels in patients presenting at less than six hours after the traumatic injury might suggest different etiologies of the inflammatory response. Comparative analysis of the study group showed no difference between the mean CRP values at admission in survivors and non-survivors.

Even if the analysis of means or medians showed the greatest difference for the NLPR, an analysis comparing the ROC curves for all the hemogram-derived immune dysregulation markers using death at 30 days after the traumatic event as a classification variable is needed. The NLPR proved to be the most reliable predictive marker of 30-day mortality, followed by PLR, NLR and PCR (Table 2 and Figure 2). This prompted us to further investigate its potential value in the prediction of TIC phenotypes and non-fatal adverse events in blunt thoracic trauma, as well as its performance as a predictor of mortality in blunt thoracic trauma patients

We found a bimodal distribution of NLPR values in deceased patients. As can be seen, there is a group of patients with extremely poor survival but who have low NLPRs (Table 1, Figure 3).

Because early deaths occur primarily due to massive bleeding and TIC, we created an ROC curve that uses death in the first 72 h as the outcome. To reduce the bias of survivors, we also included patients who had received massive transfusions in this category (Figure 4).

The optimal NLPR cut-off for predicting early deaths and the need for massive transfusions was 3.1 (sensitivity = 80.00% and specificity = 71.05%). After excluding patients with an NLPR below 3.1 (early deaths and massive transfusion), we performed another ROC curve with the remaining patients, this time using 30-day survival as a cut-off point. Thus, we established an NLPR cut-off point of 9.5. For patients with NLPR values above 9.5, we can predict the probability of dying with 84.21% sensitivity and 95% specificity (Figure 5).

A comparison of survival curves shows the highest mortality in patients presenting a hyperinflammatory status. Patients with an NLPR of between 3.1 and 9.5 have the best survival, followed by patients with an NLPR lower than 3.1. Patients with blunt chest trauma with an NLPR score greater than 9.5 had the highest chance of death (log–rank Mantel–Cox test)–Chi-Squared = 30.49, *p* < 0.0001 (Figure 6).

Patients showing a hypoinflammatory status (NLPR < 3.1) have the highest probability of death in the first days after suffering a thoracic trauma. Patients with an NLPR between 3.1 and 9.5 closely follow this curve without exceeding it at any point. Patients with a hyperinflammatory status (NLPR > 9.5) appear to have continuously increasing mortality during hospitalization. Starting from day eight, the mortality curve diverges from the other two groups. At day 30, patients with a hyperinflammatory status have an almost 80% probability of death. Based on the inflammatory response, we considered patients with an NLPR score < 3.1 as having a low inflammatory response, those with NLPR scores between 3.1 and 9.5 as having an intermediate inflammatory response, and those with NLPR > 9.5 as having a high inflammatory response, respectively. The analysis of the absolute number of patients who survived and died in each group reinforces the results of the survival curve analysis and highlights a U-shaped distribution of death (Figure 7).

The comparative analysis of the three groups shows similar ages and BMI values. The most acidotic patients were those in the high-inflammatory-response group, while the other two groups were similar in this regard. The lactate and base excess levels follow the same trend of metabolic acidosis in the three groups. The lowest median value of hemoglobin and platelets and the highest median values of the INR and aPTT were revealed in the group of patients with NLPR scores below 3.1 (Table 3).

The viscoelastic assay analysis unveiled all the specific TIC phenotypes in all three groups, ranging from hypocoagulopathy (hyperfibrinolysis, fibrinogen deficiency, thrombin-generation deficiency) to hypercoagulability (fibrinolysis shutdown, hyperfibrinogenemia, and increased thrombin generation). The low-inflammatory response group was mainly characterized by hypocoagulopathy (hyperfibrinolysis, *n* = 10 (37.04%); and fibrinogen deficiency *n* = 19 (70.37%)). The group with a high-inflammatory response showed mostly hypercoagulable traits (fibrinolysis shutdown, *n* = 14 (63.64%); and increased thrombin generation, *n* = 11 (50.00%)). However, we could identify patients with high inflammation and hypocoagulability. Patients with an intermediate-inflammatory response presented both hyper- and hypocoagulable phenotypes. However, a comparative analysis of the distribution frequencies also shows a pattern of intermediate response in terms of coagulation, being less hypocoagulable than the low-inflammation and less hypercoagulable than the high-inflammation groups. Tranexamic acid was used the least in the high-inflammatory-response group. Nevertheless, the analysis of the distribution frequency for the three studied lots does not reveal a statistically significant difference (Table 4). The hyperinflammatory group seems to have a greater platelet contribution to the blood clot, followed by the group with moderate inflammation and the low inflammatory group. However, the Kruskal–Wallis analysis does not reveal a statistically significant difference.

The high-inflammation group developed ARDS during hospitalization in the intensive care unit (*n* = 10, 45.45%) more frequently compared with the group of low-inflammation response (*n* = 4, 14.81%) and the group of patients with moderate inflammation (*n* = 8, 21.62%), *p* = 0.04; consequently, following this trend, they most often needed prolonged mechanical ventilation (defined as longer than five days). Thromboembolic events (defined as deep-vein thrombosis, pulmonary embolism, or arterial ischemia) developed during the hospital stay were also most frequent in the group of high-inflammatory-response patients. Comparative analysis of the distribution frequency of septic events (defined as any of the following: ventilator-associated pneumonia, surgical-site infections, catheter-related bloodstream infections, or catheter-associated urinary tract infections) does not show any difference among the different inflammatory response groups (Table 5).

## 4. Discussion

Trauma is characterized as a disease of an excessive innate immune response. Following traumatic injury, platelets respond to innate immune signaling and are recognized as crucial factors and damage-recognition agents with roles in hemostasis, thrombosis, and inflammation [15,16,41]. Among the validated biomarkers for the screening, stratification, and prognosis of clinical outcomes, the NLR and PLR have brought deep insights into the complex course of the inflammatory response as a consequence of the innate and adaptative immune systems and of coagulation interactions during different pathological conditions, including trauma. The NLR reflects the cellular immune response to insults ranging from the physiological response aimed at local tissue repair and recovery to hyperinflammation, prolonged inflammation, and immunosuppression, with opposite changes in neutrophil and lymphocyte counts, that is, neutrophilia and lymphopenia [42]. Thus, a high NLR reflects multifactorial and dynamic processes depending on the regulation of various immunologic, neuroendocrine, and humoral reactions such as cell margination, programmed cell death, the effect of stress hormones, and sympathetic and parasympathetic nervous systems, leading to a shift from an adaptative to an innate immunity response [42,43]. The study of platelet function after severe injury has led to an apparent paradox, whereby the platelet’s initial vital hemostatic function may be followed by late morbidity in the trauma patient as inflammatory and thrombotic sequelae evolve after injury [15]. Thus, the PLR has emerged as a tool for reflecting shifts in platelet and lymphocyte counts due to acute inflammatory and prothrombotic states. The combined NLR and PLR marker, NLPR, reflects the same physiopathological events behind the NLR and PLR but with higher sensitivity and specificity.

In our study, we found a U-shaped distribution of mortality, with high rates of early deaths in patients with an NLPR < 3.1 and high rates of late deaths in patients with an NLPR > 9.5. Both insufficient and exaggerated immune responses prove to be maladaptative. It is reasonable to hypothesize that the former group would benefit from stimulation of the immune response to match the acute stress condition adequately. In contrast, the latter would benefit from inhibition of the immune response. Before any therapeutic recommendations can be made, prospective studies should be conducted to validate the NLPR, and different immunomodulatory therapeutical interventions should be tested in vitro and in vivo.

Activation of coagulation during inflammation is a physiologic response that helps contain inflammatory activity at the site of injury and that promotes reparatory mechanisms [44]. The bidirectional relation between inflammation and coagulation is uniquely described in thoracic trauma patients due to the role taken by the lungs in both systems under normal conditions and after local injury. We found that thoracic trauma patients expressing moderate inflammation and inflammation-induced local hypercoagulation objectified as an NLPR between 3.1 and 9.5 may have a survival benefit.

When investigating NLPR’s relation with classic TIC patterns, as characterized by ROTEM analysis, we found a tendency toward hypocoagulability in the low-inflammation group and toward hypercoagulability in the high-inflammation group, respectively. The intermediate-inflammatory-response group showed an intermediate coagulation status, with less hypocoagulability than the former and less hypercoagulability than the latter. However, no relation between the inflammatory response and specific TIC patterns could be demonstrated. To our understanding, this underscores the irreplaceable role of ROTEM analysis in the precise delineation of the early stages of TIC. Specific laboratory definition of late hypercoagulable TIC remains problematic, yet several studies have identified increased clot strength and fibrinolysis shutdown, as measured by viscoelastic assays, as risk factors for thromboembolic complications [7].

Tissue factor is considered the primary initiator of coagulation during SIRS [45]. A particular interaction between inflammation and coagulation is described in tissues with endogenous expression and regulation of TF, such as brain and lung [46]. Thrombin, a product of TF-induced coagulation, is a critical participant in the crosstalk between coagulation and inflammation. It activates numerous cell types, including platelets and endothelial and immune cells, via proteolytic cleavage of protease-activated receptors (PAR 1–4). Thrombin-induced PAR activation leads to platelet activation and aggregation (thus promoting thrombosis), activates PAR-1 on endothelial cells and fibroblasts, and determines the production of proinflammatory interleukins [47].

Under normal conditions, TF is present in small amounts in the lung epithelium, vessel adventitia, and alveolar macrophages; however, its levels increase more than 10-fold during inflammation [48]. Hypercoagulation and inhibition of fibrinolysis have been described in ARDS of multiple etiologies, including sepsis, lung reperfusion injury, burn injury, and trauma [48,49,50]. The mechanisms that contribute to pulmonary coagulopathy are localized TF-mediated thrombin generation, impaired activity of natural coagulation inhibitors, activated protein C, antithrombin, and tissue factor pathway inhibitors, and the depression of tissue-type and urokinase-type plasminogen activator-mediated fibrinolysis caused by the increase in plasminogen activator inhibitor PAI-1 [51].

A French study showed that NLR levels at admission failed to predict the development of ARDS over the first five days after injury in blunt chest trauma patients. However, the NLR was more specific in predicting non-focal ARDS than focal ARDS. Focal ARDS was defined as per the Berlin criteria, with the presence of bilateral opacities on chest imaging and hypoxia not fully explained by cardiac failure or fluid overload, yet with consolidations confined to the lower and back parts of the lungs. Non-focal ARDS was defined as the diffuse loss of aeration. An NLR ≥ 15 at admission was the best threshold for the prediction of non-focal ARDS (sensitivity 0.50 [95%CI: 0.28–0.72] and specificity 0.86 [95%CI: 0.81–0.91]) in blunt thoracic trauma patients [28]. In our study, the relation between a high-inflammatory response (NLPR > 9.5) and ARDS (*p* = 0.04) could be explained by the identification of hypercoagulability as a contributor to the complex physiopathology of ARDS development. Different ARDS phenotypes in blunt thoracic trauma need to be considered in clinical practice and in further studies on personalized ventilation strategies. The NLPR needs to be analyzed in connection with previously validated ROTEM-derived markers of hypercoagulation in both ARDS and non-ARDS patients, as well as in relation to ARDS phenotypes.

Moore et al. analyzed the spectrum of post-injury fibrinolysis and its relevance to antifibrinolytic therapy in trauma patients. They found a similar U-shaped distribution of mortality, with the lowest rate in the physiologic fibrinolysis group and the highest mortality rates at the extremes of shutdown and hyperfibrinolysis (*p* < 0.0001). They also found that modest levels of fibrinolysis seem protective, compared with extremes of fibrinolysis, and that hyperfibrinolytic patients die of exsanguination early after injury. In contrast, shutdown patients more frequently have delayed mortality from organ failure [38,52].

Studies on patients who undergo elective cardiopulmonary bypass surgery and primary total knee arthroplasty have shown that TxA use might be associated with unintended positive effects on the immune system, such as inflammatory response attenuation [53,54]. Murine models of trauma, especially TBI, showed that TxA might be beneficial in modulating the inflammatory and immune response after TBI in the absence of systemic hyperfibrinolysis [55]. The massive administration of TxA in multiple trauma patients, as per international guidelines based on large, randomized, placebo-controlled trials, has shown survival benefits related to the TIC amendment [56,57]. Further studies are needed to investigate whether immune modulation independently contributes to TxA-related survival improvement and whether TxA administration as an immune modulator could be indicated in the absence of hyperfibrinolysis in trauma patients and, more specifically, in blunt chest trauma patients. Our study could not find a relation between prehospital TxA administration and inflammatory profiles. However, we consider that targeted recommendations for TxA use as an immune modulator can only be formulated after future specifically designated, prospective, randomized trials evaluating its efficacy, optimal patient population (in terms of coagulation and immune-inflammatory status), optimal dose, and administration regimen, keeping in mind clinical judgment at the core of any treatment. Be that the case, we reiterate the subsequent need to find adequate laboratory parameters that can aid the clinician in making real-life, timely medical decisions.

This study has several limitations. We included a relatively small number of patients from a single trauma center, and selection bias could not be avoided, even if we excluded patients transferred from other centers if the transfer was made later than six hours after the traumatic injury. We could not measure serum levels of pro- and anti-inflammatory cytokines and analyze them in relation to the NLPR. However, we believe that the validation of the NLPR against classic laboratory parameters of inflammation, similar to the validation of the NLR, holds great promise for future research in the field of personalized medicine [42]. It is still unknown whether the early identification of specific biomarker-defined subgroups and the development of designated care bundles might address preventable blunt thoracic trauma-related deaths.

## 5. Conclusions

Blunt thoracic trauma presents unique physiopathological traits due to the complex interaction of immune and coagulation systems in normal lung tissue and their (mal)adaptative changes following thoracic trauma. Readily available at the bedside, the inexpensive hemogram-derived marker NLPR has a predictive value for mortality in patients with extreme inflammatory responses to thoracic trauma. Patients expressing moderate inflammation, objectified as an NLPR between 3.1 and 9.5, may have a survival benefit.

## Figures and Tables

**Figure 1 jpm-14-01168-f001:**
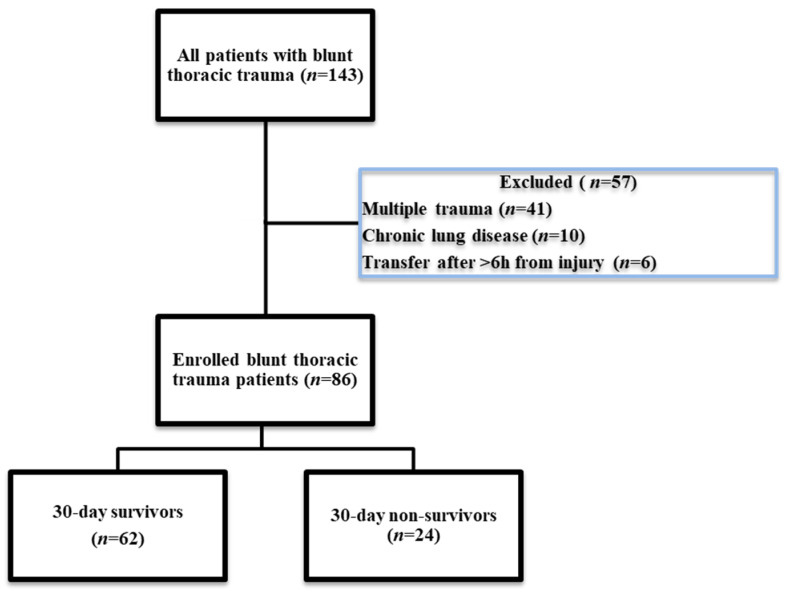
Flow diagram of patient enrollment.

**Figure 2 jpm-14-01168-f002:**
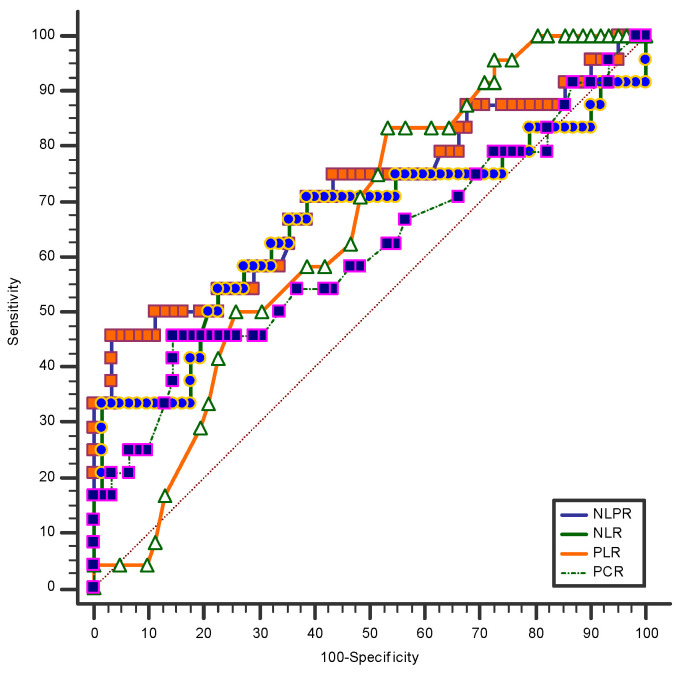
ROC curves of the NLPR for mortality prediction in thoracic trauma patients. CRP—C-reactive protein; NLR—neutrophil-to-lymphocyte ratio; PLR—platelet-to-lymphocyte ratio; NLPR—neutrophil-to-lymphocyte × platelet ratio.

**Figure 3 jpm-14-01168-f003:**
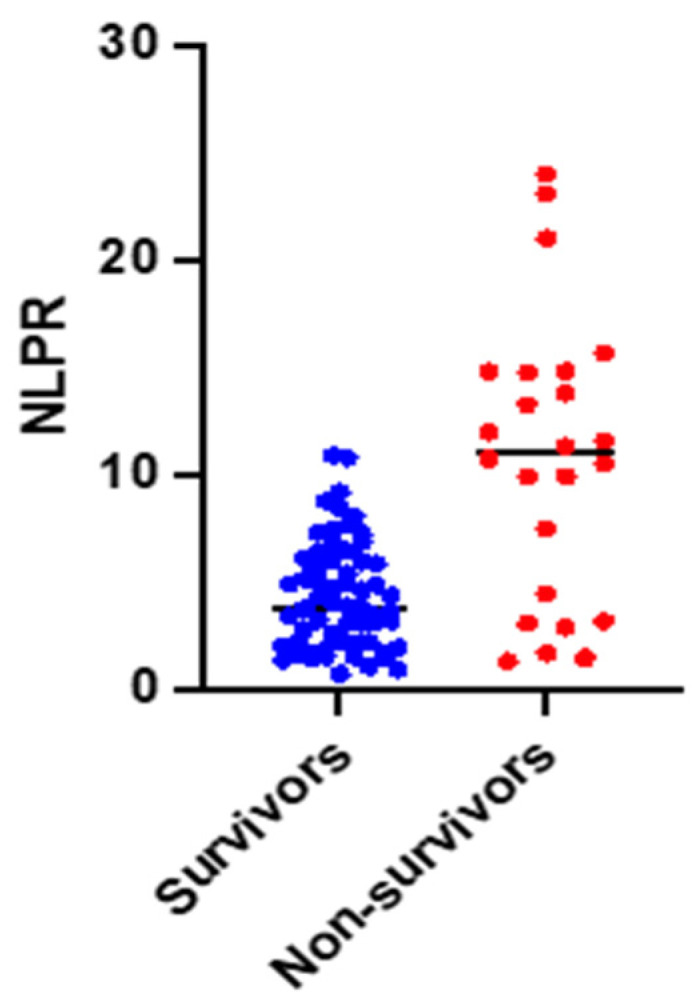
Estimation plot of the NLPR (survivors vs. non-survivors with an unpaired *t*-test).

**Figure 4 jpm-14-01168-f004:**
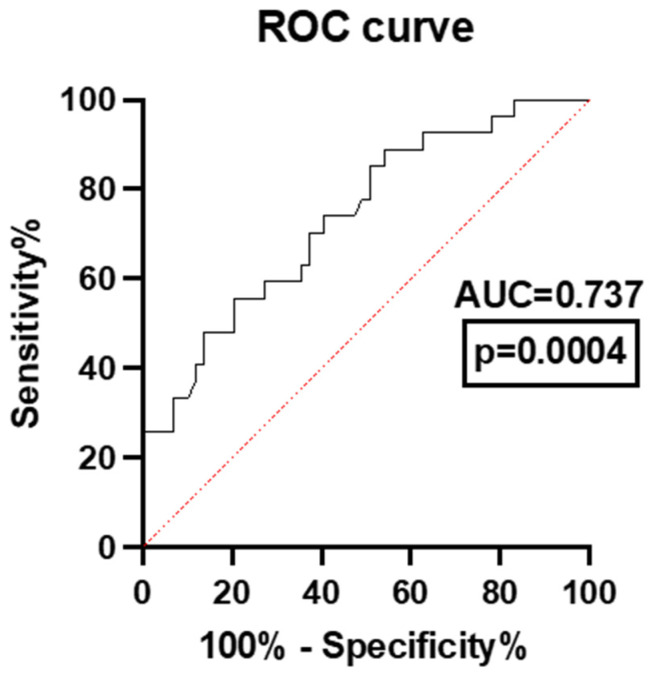
ROC curve of the NLPR for predicting early mortality and the need for massive transfusions in thoracic trauma patients.

**Figure 5 jpm-14-01168-f005:**
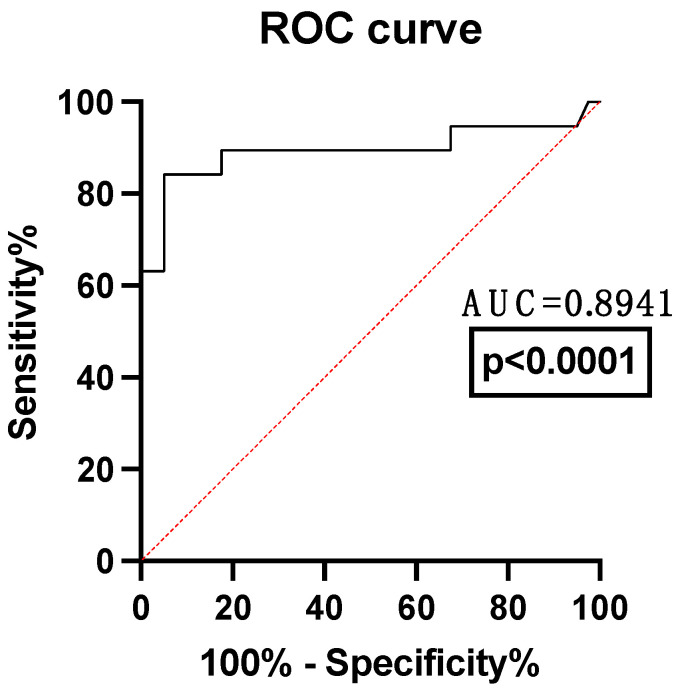
ROC curve of the NLPR for 30-day survival prediction in thoracic trauma patients.

**Figure 6 jpm-14-01168-f006:**
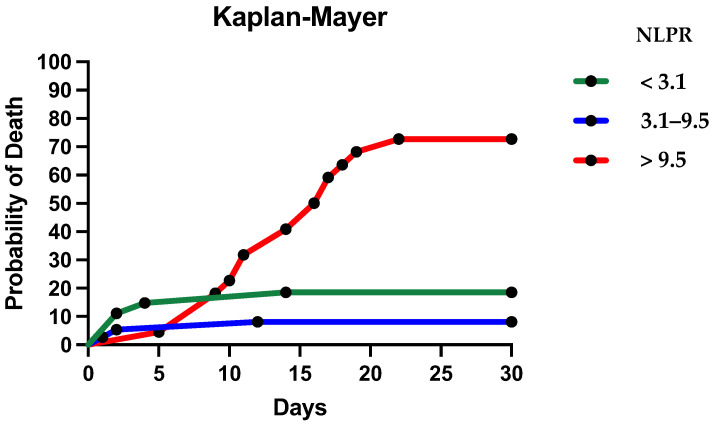
Survival curves of different phenotypes of inflammatory status based on the NLPR.

**Figure 7 jpm-14-01168-f007:**
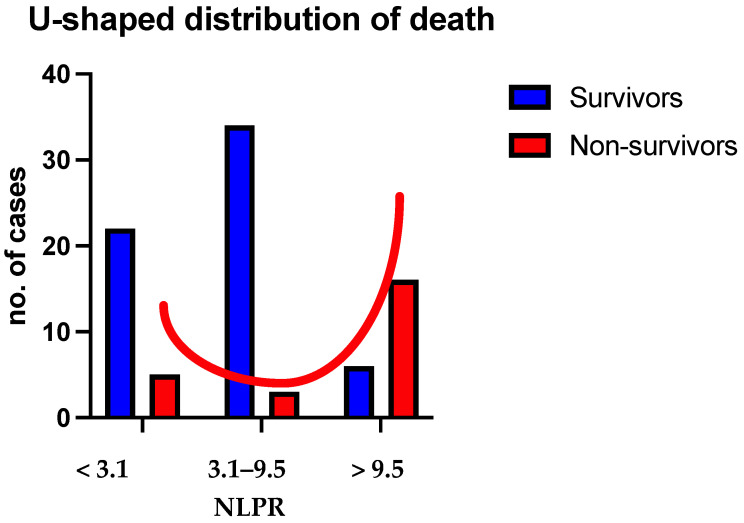
U-shaped distribution of mortality at 30 days that is related to the inflammatory status based on the NLPR.

**Table 1 jpm-14-01168-t001:** Comparative analysis between the paraclinical data of survivors and non-survivors.

Parameters	Survivors (*n* = 62)	Non-Survivors (*n* = 24)	*p* Value—Two-Tailed
Age (years) *	41.36 ± 12.86	44.86 ± 9.60	0.23
BMI (kg/m^2^) *	25.15 ± 4.21	26.11 ± 3.90	0.33
Mean chest AIS (pt) **	4 (3–4)	4 (3–4)	0.46
Surgical intervention ***	*n* = 16 (25.81%)	*n* = 8 (33.33%)	0.48
CRP (mg/L) *	2.90 (1.37 to 4.67)	3.75 (1.75 to 10,60)	0.11
NLR *	7.86 ± 3.48	10.78 ± 5.61	0.004
PLR *	0.14 (0.10 to 0.19)	0.19 (0.13 to 0.29)	0.03
NLPR *	4.31 ± 2.51	10.66 ± 6.63	<0.0001
pH *	7.31 (7.25 to 7.40)	7.18 (7.09 to 7.31)	<0.0001
Base excess (mmol/L) **	−6.05(−11.00 to −3.15)	−12.10 (−16.95 to −8.70)	0.0006
Serum lactate(mmol/L) **	2.30 (1.10 to 4.20)	4.1 (2.45 to 5.65)	0.01
Serum bicarbonate (mmol/L) *	20.23 ± 3.83	19.21 ± 5.05	0.31
Hemoglobin (g/dL) **	9.40 (7.10 to 11.10)	9.15 (6.60 to 9.55)	0.02
Platelets/mL **	188.50 (161.00 to 224.00)	103.50 (86.00 to 149.50)	<0.0001
INR **	1.66 (1.29 to 1.82)	1.95 (1.47 to 2.60)	0.003
aPTT (s) *	36.18 ± 7.23	39.46 ± 6.90	0.059
EXTEM CT (s) *	66.27 ± 14.95	63.25 ± 21.11	0.45
EXTEM A5 (mm) **	38.00 (36.00 to 40.00)	30.00 (28.00 to 33.00)	<0.0001
EXTEM MCF (mm) **	61.00 (58.00 to 62.00)	51.00 (46.00 to 54.00)	<0.0001
EXTEM ML (%) **	3.00 (0.00 to 6.00)	8.00 (2.00 to 16.00)	0.02
FIBTEM A5 (mm) **	10.00 (8.00 to 11.00)	7.00 (5.00 to 9.00)	<0.0001
FIBTEM MCF (mm) **	14.00 (12.00 to 16.00)	11.00 (7.00 to 12.00)	0.0003
FIBTEM ML (%) **	0.00 (0.00 to 3.00)	0.00 (0.00 to 7.00)	0.31
PLTEM **	138.80 (123.80 to 147.50)	94.45 (74.56 to 103.80)	<0.0001

* Unpaired *t*-test—parametrical data; ** Mann–Whitney test—non-parametrical data; *** Fischer’s exact test—the medians, means, interquartile ranges, and standard deviations are presented in the table; AIS—abbreviated injury scale; BMI—body mass index; CRP—C-reactive protein; NLR—neutrophil-to-lymphocyte ratio; PLR—platelet-to-lymphocyte ratio; NLPR—neutrophil-to-lymphocyte x platelet ratio; INR—international normalized ratio; aPTT—activated partial thromboplastin time; EXTEM CT—clotting time in the EXTEM channel; EXTEM A5—clot amplitude at 5 min/EXTEM; EXTEM MCF—maximum clot firmness/EXTEM; EXTEM ML—maximum lysis/EXTEM; FIBTEM A5—clot amplitude at 5 min/FIBTEM; FIBTEM MCF—maximum clot firmness/FIBTEM; FIBTEM ML—maximum lysis/FIBTEM; PLTEM—qualitative platelet contribution to clot formation.

**Table 2 jpm-14-01168-t002:** ROC curves of the NLPR for mortality prediction in thoracic trauma patients. AUC—area under the curve; CI—confidence interval; CRP—C-reactive protein; NLR—neutrophil-to-lymphocyte ratio; PLR—platelet-to-lymphocyte ratio; NLPR—neutrophil-to-lymphocyte × platelet ratio.

Variable	AUC	Standard Error	95%CI	*p* Value
NLPR	0.71	0.06	0.60 to 0.80	0.002
NLR	0.65	0.07	0.54 to 0.75	0.04
PLR	0.64	0.06	0.53 to 0.74	0.01
CRP	0.61	0.07	0.50 to 0.71	0.13

**Table 3 jpm-14-01168-t003:** Comparative analysis of paraclinical data between the inflammation groups objectified by the NLPR.

Parameters	NLPR			
	<3.1 (*n* = 27)	3.1–9.5 (*n* = 37)	>9.5 (*n* = 22)	*p* Value—Two-Tailed
Age (years) *	42.26 ± 10.31	41.11 ± 8.62	44.19 ± 9.12	0.47
BMI (kg/m^2^) *	25.11 ± 3.11	24.12 ± 4.82	26.31 ± 4.10	0.15
pH *	7.31 (7.27 to 7.36)	7.32 (7.18 to 7.40)	7.16 (7.09 to 7.26)	<0.0001
Base excess (mmol/L) **	−9.9 (−13.6 to −4.70)	−4.00 (−7.60 to −2.40)	−11.75 (−17.65 to −9.35)	<0.0001
Serum lactate(mmol/L) *	2.73 ± 1.52	2.58 ± 1.81	4.29 ± 1.93	0.001
Serum bicarbonate (mmol/L) *	20.48 ± 3.91	20.19 ± 3.85	18.86 ± 5.03	0.36
Hemoglobin (g/dL) **	8.70 (8.20 to 9.40)	9.40 (8.90 to 11.60)	9.70 (9.27 to 10.50)	0.0008
Platelets/mL **	134.50 (98.25 to 171.50)	183.00 (151.00 to 189.00)	218.00 (187.00 to 368.00)	<0.0001
INR **	1.96 (1.73 to 2.60)	1.37 (1.22 to 1.73)	1.69 (1.30 to 1.88)	<0.0001
aPTT (s) *	40.74 ± 7.93	34.86 ± 5.51	38.32 ± 7.28	0.02

* Differences between groups were identified with the one-way ANOVA test for parametrical data; ** Differences between groups were identified with the Kruskal–Wallis test for non-parametrical data; BMI—body mass index; NLPR—neutrophil-to-lymphocyte × platelet ratio; INR—international normalized ratio; aPTT—activated partial thromboplastin time.

**Table 4 jpm-14-01168-t004:** Comparative analysis of the distribution frequencies of the main TIC phenotypes between inflammation groups objectified by the NLPR.

	Parameters	NLPR			
		<3.1 (*n* = 27)	3.1–9.5 (*n* = 37)	>9.5 (*n* = 22)	*p* Value—Two-Tailed
Hypocoagulability	Prehospital tranexamic acid	*n* = 12 (44.44%)	*n* = 17 (47.22%)	*n* = 7 (19.44%)	0.57
	Hyperfibrinolysis *	*n* = 10 (37.04%)	*n* = 4 (10.81%)	*n* = 3 (13.64%)	0.03
	Fibrinogen deficit *	*n* = 19 (70.37%)	*n* = 9 (24.32%)	*n* = 8 (36.36%)	0.001
	Thrombin-generation deficit *	*n* = 8 (29.63%)	*n* = 6 (16.22%)	*n* = 5 (22.73%)	0.43
Hypercoagulability	Fibrinolysis shutdown	*n* = 5 (18.52%)	*n* = 11 (29.73%)	*n* = 14 (63.64%)	0.003
	Hyperfibrinogenemia	*n* = 2 (7.41%)	*n* = 4 (10.81%)	*n* = 5 (22.73%)	0.29
	Increased thrombin generation	*n* = 2 (7.41%)	*n* = 7 (18.92%)	*n* = 11 (50.00%)	0.002
Platelet contribution (PLTEM) **		102.70 (85.25 to 172.90)	132.80 (103.80 to 149.50)	138.80 (118.90 to 146.80)	0.12

* Differences between distribution frequencies were identified with the Fischer’s exact test. ** Differences between groups were identified with the Kruskal–Wallis test for non-parametrical data, Hypocoagulopathy: hyperfibrinolysis is defined as ML/EXTEM > 15% or ML/FIBTEM > 10%; fibrinogen deficit is defined as A5/EXTEM < 35 mm or A5/FIBTEM < 9 mm or MCF/FIBTEM < 12 mm; and thrombin-generation deficit is defined as CT/EXTEM > 80 s or CT/INTEM > 240 s. Hypercoagulability: fibrinolysis shutdown is defined as ML/EXTEM < 3%; hyperfibrinogenemia is defined as MCF/FIBTEM > 22 mmm; and increased thrombin generation is defined as CT/EXTEM < 45 s or MCF/EXTEM > 68 mm. CT—clotting time; A5—maximum clot amplitude at 5 min; MCF—maximum clot firmness; ML—maximum lysis.

**Table 5 jpm-14-01168-t005:** Comparative analysis of the distribution frequencies of primary outcomes between inflammation groups objectified by the NLPR.

Parameters	NLPR			
	<3.1 (*n* = 27)	3.1–9.5 (*n* = 37)	>9.5 (*n* = 22)	*p* Value—Two-Tailed
ARDS *	*n* = 4 (14.81%)	*n* = 8 (21.62%)	*n* = 10 (45.45%)	0.04
Prolonged mechanical ventilation *	*n* = 8 (29.63%)	*n* = 16 (45.71%)	*n* = 11 (50.00%)	0.001
Thrombotic events *	*n* = 2 (7.41%)	*n* = 5 (13.51%)	*n* = 8 (36.36%)	0.03
Septic events *	*n* = 6 (22.22%)	*n* = 8 (21.62%)	*n* = 10 (45.45%)	0.12

* Differences between distribution frequencies were identified with the Fischer’s exact test.

## Data Availability

The original contributions presented in this study are included in the article. Further inquiries can be directed to the corresponding authors.

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
