# Peer review of "Hemogram-Based Phenotypes of the Immune Response and Coagulopathy in Blunt Thoracic Trauma"

_jpm, 2024, doi:10.3390/jpm14121168_

Round 1
Reviewer 1 Report
Comments and Suggestions for Authors
Interesting observational work although retrospective and in a relatively small cohort of patients. The findings are very relevant and in keeping with the hypothesis of inflammation and coagulation being closely linked.
I would caution the authors to not make any conclusions from this study and rather prompt a future prospective, observational study to test the hypothesis. Also, any therapeutic intervention is still lacking to target the immune response and unfortunately, these findings do not change management of these patients.
The theories around the use of Tranexamic acid is interesting and warrants further work.
Author Response
Concerning the revised version of manuscript no. 3196135 entitled “Hemogram-based phenotypes of immune response and coagulopathy in blunt thoracic trauma”, we appreciate the time and effort dedicated to peer-reviewing and proofreading our manuscript. The feedback and suggestions provided not only enhanced the quality of our paper but also offered valuable insights for our future work.
Thank you for your kind remarks. Indeed, our study is limited by the retrospective design and the analysis of data from a relatively small number of patients coming from a single trauma center. The controversial use of tranexamic acid as an immune modulator warrants future research and complex study methodology involving a deeper dive into the molecular mechanisms of action in different settings, and more importantly, in various moments in a trauma patient’s evolution, since it is proven, but not completely understood why tranexamic acid use is time-sensitive. We completely agree that no clinically-translatable conclusions should be made based on our study only and we highly encourage future research on this topic.
Reviewer 2 Report
Comments and Suggestions for Authors
The authors present a retrospective study of hemogram-derived ratios from patients with blunt thoracic trauma. The results of this limited cohort study show great promise to inform treatments and condition management without additional invasive investigations.
The manuscript has been well written, provides a strong background to the topic, and holds great value to the research area, not only the specific thoracic case but more broadly.
Author Response
Concerning the revised version of manuscript no. 3196135 entitled “Hemogram-based phenotypes of immune response and coagulopathy in blunt thoracic trauma”, we appreciate the time and effort dedicated to peer-reviewing and proofreading our manuscript. The feedback and suggestions provided not only enhanced the quality of our paper but also offered valuable insights for our future work.
With reference to the comments and suggestions:
Thank you for your kind words. As you have very well underscored, the main limitation of our study, that is the retrospective design and the analysis of data from a relatively small number of patients coming from a single trauma center, serves as a useful starting point for future research in larger, more complex groups of trauma patients.
We aim to support future research for the early identification of specific biomarker-defined subgroups and to help develop targeted care bundles as well as precision, personalized care for trauma patients.
Reviewer 3 Report
Comments and Suggestions for Authors
Dear authors, first of all I want to congratulate for your excellence work!! This is a very interesting issue and you have described it in an excellent way! The methods are clearly explained and the results are clarified in the most accurate way. The tables and figures are very helpful for the readers
Maybe a comparison of you results with other similar studies to increase the significance of your study, although this is out of the field of this study.
Author Response
Dear Editor,
Concerning the revised version of manuscript no. 3196135 entitled “Hemogram-based phenotypes of immune response and coagulopathy in blunt thoracic trauma”, we appreciate the time and effort dedicated to peer-reviewing and proofreading our manuscript. The feedback and suggestions provided by the reviewers not only enhanced the quality of our paper but also offered valuable insights for our future work.
With reference to the comments and suggestions:
Thank you for your kind remarks. Comparing results between similar studies on this topic was of great interest to our group since we acknowledge its contribution to expanding the knowledge of the topic, deepening the understanding of the physiological mechanisms behind coagulation and inflammation, as well as increasing the significance of the results of our study. We could not find a similar study specifically targeting blunt thoracic trauma patients and we consider this both a proof of originality and a limitation of our study.
Reviewer 4 Report
Comments and Suggestions for Authors
Thankyou for the opportunity to review your manuscript.
The content is interesting and findings provide additional innovative avenues for research.
Abstract. the first sentence of results lacks any statistical value.
Introduction. Provides a good background although the sentence 'We hypothesized that thoracic trauma patients exhibit different patterns of coagulation and inflammation abnormalities from the early stages' requires more justification as this is the cohort that you will be examining.
Materials. 2.1 Figure 1 - n=143 - is that all thoracic trauma or just blunt thoracic trauma? Can you please outline inclusion and exclusion criteria explicitly It is also not clear which blood results you are basing the study off - what is meant by 'blood work analysis performed at admission...' is this the first set of bloods on arrival or the worst set of bloods during their resuscitation in the ED? If its the first set of bloods, why have you chosen this time point given that, especially for AIS 4,5, their blood work will likely worsen 2.2 Data collection - what software was used to collect the data for analysis eg redcap etc? A lot of 2.2 should be in 2.3 as its interpreting the investigation findings rather than data collection per se For 2.4 (and some of 2.3) there's a lot of descriptions around clinical care and testing which is not necessary (eg model of ventilator used and ventilator settings etc).
Results. Table 1. Mean AIS would also be useful to have. Very comprehensive results discussion around blood measurements, many of which would be more suitable for an appendix or supplement rather than within the main text.
Discussion. The first two paragraphs should go in the introduction or removed as they are not part of the discussion of the project. A good discussion otherwise. Given the conclusion poses the question around care bundles, some discussion around this with literature reference would be useful, otherwise that point should be removed in the conclusion.
Conclusion. highly relevant
Comments on the Quality of English LanguageSmall typographical and grammatical errors. Please check thoroughly.
Author Response
We are pleased to resubmit for publication the revised version of manuscript no. 3196135 entitled “Hemogram-based phenotypes of immune response and coagulopathy in blunt thoracic trauma”. We appreciate the time and efforts made in proofreading this manuscript. With reference to the comments and suggestions:
Abstract. the first sentence of results lacks any statistical value.
We decided to eliminate the first sentence from the abstract. The idea is later discussed and supported by evidence in the introduction.
Introduction. Provides a good background although the sentence 'We hypothesized that thoracic trauma patients exhibit different patterns of coagulation and inflammation abnormalities from the early stages' requires more justification as this is the cohort that you will be examining.
We made a change to the working hypothesis, replacing “different patterns” with “clinically relevant patterns” for a clearer direction for our research interest. The discussion section thoroughly presents the physiopathologic pathways that support the hypothesis.
Materials. 2.1 Figure 1 - n=143 - is that all thoracic trauma or just blunt thoracic trauma? Can you please outline inclusion and exclusion criteria explicitly It is also not clear which blood results you are basing the study off - what is meant by 'blood work analysis performed at admission...' is this the first set of bloods on arrival or the worst set of bloods during their resuscitation in the ED? If its the first set of bloods, why have you chosen this time point given that, especially for AIS 4,5, their blood work will likely worsen 2.2 Data collection - what software was used to collect the data for analysis eg redcap etc? A lot of 2.2 should be in 2.3 as its interpreting the investigation findings rather than data collection per se For 2.4 (and some of 2.3) there's a lot of descriptions around clinical care and testing which is not necessary (eg model of ventilator used and ventilator settings etc).
At your very welcomed suggestion, we specifically added to both the text and the flow chart diagram the word “blunt”. We identified n=143 patients with blunt thoracic trauma and enrolled n=86 patients with isolated blunt thoracic trauma in our study. We also mentioned that the blood samples that were analyzed were the first to be drawn, right at the admission of the patient. First of all, we wanted to test our physiopathological hypothesis, that signs of deranged coagulation-inflammation relationship are identifiable from admission, and secondly, we wanted to investigate the earliest possible data that could serve for prognostication. All data was collected manually and stored in Microsoft Office Excel for Mac, version 2021. GraphPad version 10.2 software was used for statistical analysis.
Indeed, points 2.2 and 2.3 were a bit hard to differentiate, thus we rearranged them under a single 2.2 point that comprises all the technical information.
We considered the detailed mentions around clinical care and testing justified due to their possible conflicting role in the evolution of thoracic trauma patients. Mode of ventilation, respiratory interventions, and fluid balance, all play a part in the modulation of local lung inflammation. It was far beyond the scope of our study to quantify this role, but we had to mention these factors, should our study be replicated.